# Study Protocol for a Pilot, Open-Label, Prospective, and Observational Study to Evaluate the Pharmacokinetics of Drugs Administered to Patients during Extracorporeal Circulation; Potential of In Vivo Cytochrome P450 Phenotyping to Optimise Pharmacotherapy

**DOI:** 10.3390/mps2020038

**Published:** 2019-05-13

**Authors:** Santosh Kumar Sreevatsav Adiraju, Kiran Shekar, Peter Tesar, Rishendran Naidoo, Ivan Rapchuk, Stephen Belz, John F Fraser, Maree T Smith, Sussan Ghassabian

**Affiliations:** 1Centre for Integrated Preclinical Drug Development, School of Biomedical Sciences, Faculty of Medicine, University of Queensland, 4072 Brisbane, Australia; maree.smith@uq.edu.au (M.T.S.); Susan.ghassabian@gmail.com (S.G.); 2Adult Intensive Care Services, The Prince Charles Hospital, 4032 Chermside, Australia; shekarkiran@yahoo.com (K.S.); stephen.belz@health.qld.gov.au (S.B.); john.fraser@health.qld.gov.au (J.F.F.); 3Critical Care Research Group, The Prince Charles Hospital, 4032 Chermside, Australia; rishendran.naidoo@health.qld.gov.au; 4Department of Cardiothoracic Surgery, The Prince Charles Hospital, 4032 Chermside, Australia; peter.tesar@health.qld.gov.au; 5Department of Anesthesia, The Prince Charles Hospital, 4032 Chermside, Australia; ivan.rapchuk@health.qld.gov.au; 6School of Pharmacy, Faculty of Health and Behavioral Sciences, The University of Queensland, 4072 Brisbane, Australia

**Keywords:** extracorporeal circulation, extracorporeal membrane oxygenation, cardiopulmonary bypass, cytochrome P450 (CYP), CYP phenotyping

## Abstract

Pharmacokinetic alterations of medications administered during surgeries involving cardiopulmonary bypass (CPB) and extracorporeal membrane oxygenation (ECMO) have been reported. The impact of CPB on the cytochrome P450 (CYP) enzymes’ activity is the key factor. The metabolic rates of caffeine, dextromethorphan, midazolam, omeprazole, and Losartan to the CYP-specific metabolites are validated measures of in vivo CYP 1A2, 2D6, 3A4, 2C19, and 2C9 activities, respectively. The study aim is to assess the activities of major CYPs in patients on extracorporeal circulation (EC). This is a pilot, prospective, open-label, observational study in patients undergoing surgery using EC and patients undergoing laparoscopic cholecystectomy as a control group. CYP activities will be measured on the day, and 1–2 days pre-surgery/3–4 days post-surgery (cardiac surgery and Laparoscopic cholecystectomy) and 1–2 days after starting ECMO, 1–2 weeks after starting ECMO, and 1–2 days after discontinuation from ECMO. Aforementioned CYP substrates will be administered to the patient and blood samples will be collected at 0, 1, 2, 4, and 6 h post-dose. Major CYP enzymes’ activities will be compared in each participant on the day, and before/after surgery. The CYP activities will be compared in three study groups to investigate the impact of CYPs on EC.

## 1. Introduction

More than 1 million cardiac operations using CPB are performed annually worldwide [1]. A major benefit of CPB is that it provides a bloodless heart for cardiac surgery, with blood flow diverted to an extracorporeal circuit. While CPB usually lasts a few hours, extracorporeal membrane oxygenation (ECMO), a modified application of CPB concept, provides longer-term extracorporeal life support (days to weeks) as a bridge to recovery or a bridge to transplant or a destination device [1]. Both CPB and ECMO necessitate exteriorisation of significant blood volumes to achieve extracorporeal oxygenation. This process results in significant pathophysiological alterations, not all of which are clearly understood [2].

Both CPB and ECMO may induce postoperative pathophysiological changes related to systemic inflammatory response syndrome (SIRS) and/or a multiple organ dysfunction syndrome (MODS) due to several interlinked mechanisms such as the exposure of blood to non-physiologic surfaces, coagulopathy, surgical trauma, anaesthesia, increased intestinal permeability to endotoxins, ischemia/reperfusion injury, hypotension, capillary leakage, and multiple organ injury [2,3,4,5,6,7]. Khabar et al. [3] discussed the link between the contribution of potential pathological post-CPB phenomena, including endotoxin and the pro-inflammatory cytokines, TNF-α, IL-6, and IL-8 as their plasma concentrations were increased significantly compared with their pre-CPB levels [3]. A significant increase in the ratio of pro-inflammatory cytokines (IL-6, IL-8, and TNF-α) to the major anti-inflammatory cytokine (IL-10) at seven days after CPB compared with a non-CPB control group was reported in another study [8]. The rapid rise in plasma concentrations of TNF-α and IL-8 almost immediately (within 15 min) after the initiation of ECMO suggests that these cytokines may be important mediators in the development of ECMO-related inflammation [2,7]. In an animal model, IL-6 concentrations in the lungs have been shown to increase after Veno-venous (VV) –ECMO [9]. In other work, the concentrations of the pro-inflammatory interleukins (IL), IL-1β and IL-8, increased in broncho alveolar lavage fluid during VA (Veno arterial)-ECMO treatment. In other work, the plasma concentrations of the major anti-inflammatory cytokine, IL-10, were significantly higher in the broncho alveolar lavage fluid of VV-treated animals [10]. There are several examples in the clinical literature whereby cytokines have been implicated in altering cytochrome P450 (CYP) isoenzyme activities during cardiopulmonary bypass [11] resulting in impaired metabolism of clinically important drugs leading to their accumulation in the body and potential the potential for causing adverse effects.

Therefore, pharmacokinetics (PK) and pharmacodynamics (PD) of administered drugs can be significantly affected in setting of an extracorporeal circuit. Multiple studies conducted in patients undergoing CPB have reported on altered drug disposition [12,13,14,15]. During cardiac surgeries using CPB, patients undergo major physiological changes including hypothermia, haemodilution, reduced blood flow and pressure, release of stress-reactant hormones, fluid and electrolyte shifts, as well as release of cytokines and nitric oxide. CPB institution thus has profound effects on the distribution and elimination of drugs, consequently affecting plasma drug concentrations [15]. These effects constantly change during surgeries using CPB and some continue to exert their influence after the patient has been successfully separated from CPB. Drugs that have critical roles in determining patients’ health and surgical outcomes such as anaesthetics, opioid analgesics, neuromuscular blockers, and antibiotics, are reportedly affected during and after surgery using CPB [11].

As with CPB, significant physiological changes occur in patients on ECMO, which may impact drug disposition and the pharmacological effect. ECMO alters PK by increasing the volume of distribution (Vd) through several mechanisms including drug adsorptive losses onto components of the circuit, haemodilution, and other physiological changes, and impaired drug elimination and sequestering drugs in the ECMO circuit [16]. Haemodilution from priming solutions on commencement of ECMO, ongoing blood product transfusions, sequestration of drugs to the circuit, and administration of fluids to maintain circuit flows has a greater effect on Vd of drugs whose distribution is limited to the plasma compartment; this is particularly relevant for hydrophilic drugs [16,17]. ECMO is also associated with reduced drug elimination and a decrease in the clearance of drugs, which is thought to occur by several mechanisms including renal dysfunction in ECMO patients, reduced metabolism, and reduced regional liver blood flow especially for drugs with a high hepatic extraction ratio as reviewed by Shekar et al. in 2012 [17]. Lack of response or toxicity as a result of altered PK of critically important medications during and post-extracorporeal circulation can potentially risk patients’ lives or extend the recovery process.

Although PK alterations on CPB and ECMO have been studied, there are sparse data on metabolism of administered drugs in these patients. It is possible that such profound physiologic insult will affect drug metabolism significantly. Therefore, we hypothesised that CPB and ECMO will significantly alter the activity of CYP enzymes that catalyse drug metabolism in humans. CPB and ECMO will increase pro-inflammatory cytokine concentrations and inhibit CYP activity to alter the pharmacokinetics of important drugs administered to patients in the peri-operative period.

The aim of this pilot clinical study is to explore a possible link between the inflammatory response caused by CPB or ECMO and CYP isoenzyme inhibition leading to changes in the PK of critically important medications. In this study, a well-characterised drug metabolism phenotyping cocktail comprising single low doses of substrates of 5 major CYP isoenzymes (reference to few CYP phenotyping cocktails that used the same probes), will be administered in patients undergoing treatments involving CPB or ECMO. The pilot study will provide preliminary information regarding the extent of changes and their significance in the activities of major CYP isoenzymes in patients undergoing cardiac surgery with CPB, and/or during the ECMO therapy. To gain insight on the impact of surgery (and not cardiopulmonary circulation) on the activities of CYP isoenzymes, patients undergoing heart surgery without CPB or ECMO are required; however, this is not practical in current cardiac practice. Therefore, a different patient population with similar demographic variables as a control group; viz patients undergoing laparoscopic cholecystectomy will be recruited.

## 2. Experimental Design

Using an in vivo CYP phenotyping procedure, the activities of CYP isoenzymes on the day of surgery will be compared 1–2 days before or 3–4 days after surgery involving CPB. For patients on ECMO, CYP phenotyping will be conducted at 1–2 days and 1–2 weeks after commencement of ECMO and 1–2 days after termination of ECMO therapy. Patients scheduled to undergo laparoscopic cholecystectomy will be recruited as a control group and CYP phenotyping will be conducted on the day of surgery, and 1–2 days before/3–4 days after surgery (Figure 1). The study is a prospective, open-label, observational study that will recruit patients in the CPB group, the ECMO group, and in the control group at the Prince Charles Hospital, Brisbane, QLD, Australia. Inflammatory and pro-inflammatory cytokines including IL-1β, IL-12p70, IL-10, IL-8, IL-6, IL-4, IL-2, TNF-α, IL-13, and INF-γ will be measured in plasma samples collected at time zero of each CYP phenotyping stage.

## 3. Procedure

### 3.1. Objectives

To describe the activities of CYP enzymes responsible for metabolism of clinically important drugs following CPB or ECMO support.To explore the impact of cytokine release during CPB and ECMO on the changes in the activities of CYP enzymes.

### 3.2. Participants

Informed consent will be obtained from study participants or surrogate decision makers as applicable. Eligible patients will be eligible for cardiac surgery using CPB, or ICU patients requiring ECMO, or laparoscopic cholecystectomy admitted at the Prince Charles Hospital, Brisbane, QLD, Australia. A small number of participants (20 participants in CPB and control group and 10 in ECMO group) are expected to be recruited in the study. Due to the large number of patients admitted to the hospital for cardiac surgery and laparoscopic cholecystectomy, the target numbers of participants are achievable. The recruitment of patients on ECMO may be slower due to the unpredictable attendance of participants.
**Inclusion Criteria**Age >18 years and <90 years;Eligible for elective mitral valve or aortic root surgeries involving CPB; orEligible for ECMO support; orEligible for elective laparoscopic cholecystectomy.**Exclusion Criteria**No consent;Pregnancy;Serum bilirubin >150 µmol/L;Already enrolled in an interventional ECMO-related research study (as per HREC approval);Ongoing massive blood transfusion requirement (>50% blood volume transfused in the previous 8 h);Therapeutic plasma exchange in the preceding 24 h;Adverse reaction to any elements of the study drug mixture;People with cognitive impairment or mental illness;Patients with significant coronary artery disease and/or aortic stenosis;Patients with liver disease/dysfunction;Using drugs that are known to be strong inhibitors or inducers of CYP enzymes (Flockhart table);Smokers.

### 3.3. Phenotyping Cocktail Administration

After informed consent is obtained, 1–2 days before/3–4 days after surgery, an indwelling venous catheter (18 G) will be inserted to the participant’s vein (forearm) for blood sampling. One blood sample (5 mL or one teaspoon) will be taken using the same catheter and participants will then be administered the in vivo drug metabolism phenotyping cocktail that will contain the following drugs:
Caffeine: 50 mg, Caffeine Liquid 20 mg/mL, 2.5 mL will be administered to the patient;Dextromethorphan: 30 mg, 10 mg/5 mL, 15 mL via oral dosing syringe will be administered to the patient;Midazolam: 1 mg, 5 mg/5 mL, solution for injection: 1 mL in oral dosing syringe will be administered to the patient; Losartan: 5 mg, 50 mg tablet will be crashed using a mortar and pestle. Powder will be dissolved in 10 mL of water; 1 mL (5 mg) via oral dosing syringe will be administered to the patient;Omeprazole, 20 mg, one 20 mg tablet will be administered orally to the patient.

Four extra blood samples (total volume of 14 mL) will be taken from participants at 1, 2, 4, and 6 h after phenotyping cocktail administration using the indwelling venous catheter (Figure 2). 

The same procedure (administration of phenotyping cocktail and collection of blood samples) will be repeated on the day of surgery. The control group of patients will receive the phenotyping cocktail and blood sampling will be conducted on two occasions; viz 1–2 days before/3–4 days after, and on the day of laparoscopic cholecystectomy surgery.

For ECMO patients, a peripheral arterial line will already be in place. At 1–2 days after starting ECMO therapy, one blood sample (3.5 mL) will be taken using the existing arterial line followed by the administration of the drug phenotyping cocktail. Four extra blood samplings will be conducted at 1, 2, 4, and 6 h after administration of the phenotyping cocktail. The CYP phenotyping procedure will be repeated at 1–2 weeks after commencement of ECMO therapy, and 1–2 days after termination of ECMO therapy (Figure 2).

Blood samples will be collected in Blue top tube (sodium citrate as the anticoagulant; 3.5 mL) and will be centrifuged at 3000 x g as soon as possible after collection, and plasma will be separated and transferred to pre-labelled eppendorf tubes. Plasma samples will be transferred into a freezer to be stored at –80 °C until the time of analysis. Patients will be closely monitored for unlikely adverse effects by clinical staff. Prescribed medications of the study participants will be unchanged during this pilot study.


**Data Collection and Management**


An electronic database will be maintained by the study site. The data to be collected are shown in Table 1.

For each patient, various de-identified clinical and demographic data will be collected by trained research staff onto a data collection form (Table 2 and Table 3). The original paper copy of forms will be stored securely at the investigation site. The electronic database will be protected using a password. Only chief investigators will have access to the data. 


**Sample Analysis**


Validated liquid chromatography tandem mass spectrometry (LC-MS/MS) bioanalytical methods, which have been developed previously, will be used to analyse samples [18]. All samples will be assayed alongside calibration standards and quality control samples, and assays will meet the acceptance criteria based on EMA guidelines [19]. All plasma samples will be stored in the investigation site for seven years after the completion of the study.

### 3.4. Data Analysis

The activities of CYP1A2, CYP2C19, CYP2D6, CYP3A, and CYP2C9 will be assessed as per the metrics outlined below.
CYP2D6: Ratio of area under the plasma concentration—time curve of dextrorphan from 0 to 6 h after drug administration (AUC_0–6h_) to dextromethorphan AUC_0–6h_.CYP3A: Ratio of 1′-hydroxymidazolam AUC_0-6h_ to midazolam AUC_0–6h_.CYP2C9: Ratio of E-3174 (losartan carboxy acid) AUC_0-6h_ to losartan AUC_0-6h_.CYP1A2: Ratio of paraxanthine AUC_0-6h_ to caffeine AUC_0–6h_ (µmol/L).CYP2C19: Ratio of 5-hydroxyomeprazole AUC_0-6h_ to omeprazole AUC_0–6h_.


**Adverse Events**


As low single doses of probe drugs are used in most phenotyping cocktails, this ensures patient/subject safety and minimises the potential for probe drug-drug interactions. No side effects have been reported in previous studies that used caffeine, dextromethorphan, omeprazole, midazolam, and losartan as probe drugs in CYP phenotyping cocktails [20,21].

The drugs that will be administered to patient as part of this study are either standard therapeutic doses or low single doses. Serious adverse events are unlikely given the nature of the medications administered and doses used. Serious adverse events will be reported to the reviewing HREC immediately. In addition, a summary of serious adverse events will be submitted to the reviewing HREC every six months. 


**Statistical Analysis**


As the study is of an exploratory and observational design, it may not be possible to conduct full statistical analysis. The CYP activities will be compared in each study group 1–2 days before/3–4 days after surgery and on the day of surgery and also between the CPB and ECMO groups compared with the laparoscopic cholecystectomy control group using non-parametric Mann–Whitney U test.

## 4. Ethical and Dissemination

Consent from patients or their legally authorized representatives will be obtained by experienced clinicians and all questions and concerns from patients’ family members and their caregivers will be addressed in a timely manner. Drugs that will be administered to participants as the CYP substrates have been in the market for many years and all their possible side effects regarding their multiple dosing administrations are known [20,22,23,24]. The research application was approved by the Prince Charles Hospital, Metro North Hospital and Health Service (HREC/16/QPCH/39) and the University of Queensland (Clearance no. 2016001643) Human Research Ethics Committees. After the study is completed, the database will be closed and followed by statistical analyses, interpretation of results, and dissemination to scientific journals.


**Patient and Public Involvement**


Neither patients nor public were involved in the development of the research question, study design, outcome measures, recruitment to and conduct of the study, or assessment of the burden of the intervention. The results of the study will be disseminated to the interested study participants by means of lectures given by the investigators.

## 5. Discussion

This proposed exploratory clinical study will use an in vivo phenotyping drug cocktail approach to provide key data on the extent to which CPB and/or ECMO alter drug metabolising enzyme activities in patients undergoing EC support. Lack of knowledge about how CPB/ECMO impacts the activities of important drug metabolising enzymes may lead to suboptimal pharmacotherapy and may result in therapeutic failure or drug toxicities. Significant changes, if observed in the activities of CYP isoenzymes in this pilot study, may provide justification to run a subsequent larger-cohort study with adequate power to evaluate the extent of changes on CYP activities along with other variables using PK modelling techniques.

The phenotyping cocktail approach is used to assess the effects of drugs on CYP enzyme activity in vivo and it is a validated and well-tolerated method to measure CYP activities in vivo. This phenotyping cocktail approach has been used in clinical studies such as:Assessment of the potential for metabolic drug-drug interactions of novel drugs [25,26]; Identification of the main CYPs involved in the metabolism of particular drugs of interest [27,28,29]; Assessment of changes in CYP activities in disease states such as congestive heart failure (CHF), human immunodeficiency virus (HIV)-AIDS, and various types of cancer [30,31,32,33]; Evaluation of the impact of herbal medicines and aging on major CYP enzyme activities [34,35,36,37];Enhancement of our collective understanding of the impact of drug metabolism on pharmacotherapy in psychiatry [38].

Most drug metabolism phenotyping cocktails used during the last two decades have been shown to be safe due to the low doses used in patients and also the lack of drug–drug interactions between the probe compounds used to phenotype individual CYPs [39]. One exception is the case of a severe adverse reaction in healthy female volunteers after administration of a phenotyping cocktail containing tramadol, omeprazole, losartan, and caffeine as probe drugs for CYP2D6, CYP2C19, CYP2C9, and CYP1A2, respectively [39]. These unexpected adverse reactions were attributed to tramadol and the authors warned against the combination of tramadol with omeprazole, losartan, and caffeine.

The findings from our proposed pilot study, together with the subsequent prospective larger-cohort clinical study, are anticipated to provide the data enabling development of a guideline to optimise pharmacotherapy in patients undergoing extracorporeal circulation. The number of participants to be recruited is based on the expected number of admitted patients in the hospital in the period of study. If we see high variability in the CYP activities that does not provide enough information for the calculation of the power of the subsequent study, we need to extend the study to recruit more patients. The findings from this study will be novel and invaluable in providing evidence-based dosing schedule for drugs administered during treatment involving EC to improve patient outcomes, reduce mortality, length of stay in ICU, and costs associated with the treatment of complications caused by suboptimal drug therapy. 

## 6. Conclusions

Appreciating CYP activity during EC is a key step in understanding altered PK in patients supported with CPB and ECMO life support. The knowledge generated through this pilot study may lead to development of PK models that incorporate these alterations in cytochrome P450 activity, EC life support, and critical illness-induced PK changes to generate robust drug dosing guidelines in patients undergoing cardiac surgery and/or EC life support. 

## Figures and Tables

**Figure 1 mps-02-00038-f001:**
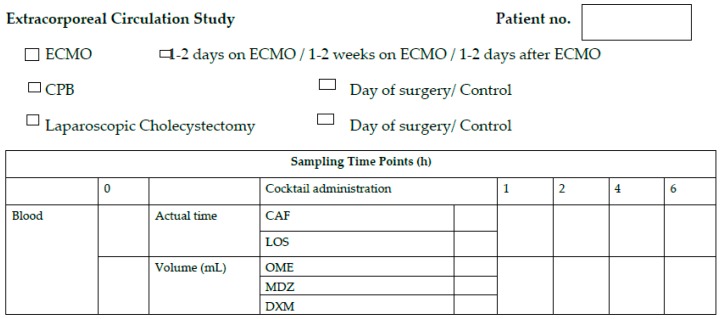
Blood Sampling Schedule.

**Figure 2 mps-02-00038-f002:**
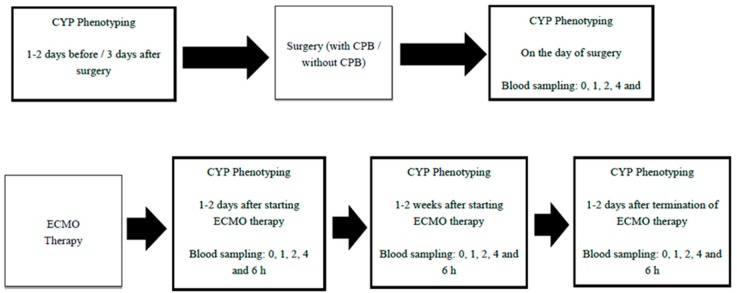
CYP phenotyping schedule for patients on ECMO and cardiac surgery with CPB.

**Table 1 mps-02-00038-t001:** Data collection and management.

Type of Data	Details Recorded
**Demographic Data**	Age Gender Weight Height Food intake Water intake
**Clinical Data**	Admission diagnosis Illness severity scores [Sequential Organ Failure Assessment (SOFA) on the day of sampling & Acute Physiology and Chronic Health Evaluation (APACHE III) on admission]. Use of renal replacement therapy for patents on ECMO Other diseases Other medications
**Organ Function Fata**	Serum bilirubin, total protein and albumin concentrations Serum creatinine concentrations 8-hour urinary creatinine clearance 24 h fluid balance and blood product requirements
**ECMO Data**	Days of therapy ECMO flows during sampling period Type of oxygenator and pump
**CPB Data**	Duration of CPB Type of CPB Type of the hemofilter, blood and effluent pump

**Table 2 mps-02-00038-t002:** Demographic and clinical collection form for patients on ECMO.

Data Collection Form Extracorporeal Circulation Study (ECMO)	Patient No.
Days on ECMO:Days off ECMO:	Date:Time:Food intake: Water intake:
Age	Sex (M/F)	Weight (Kg)	Height (cm)
ICU Admission Diagnosis	APACHE III	SOFA
Days/hours on ECMO	
ECMO flow rate	
Type of ECMO	VA 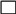 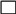 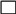 VV Other
Pump	Josta Rota flow 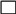 Levotronix Centrimag 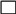 Cardiohelp 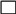
Oxygenator	QuadroxOther (please specify) ………
Serum bilirubin (µmol/L)	RRT: Yes/NoIf yes, please specify mode and flow below:
Midazolam Infusion Bolus (YES/NO)	Midazolam continuous infusion (YES/NO)
Serum Albumin (µmol/L)	CVVH	CVVHDF	SCUF
Serum Creatinine (µmol/L)	EDD	IHD	OTHER
Total Protein (g/L)	Effluent Flow Rate (mL/h)
Blood Urea (mmol/L)	Blood Flow Rate (mL/h)8h Creatinine Clearance
Blood Product Transfusion Details	24 h Fluid Balance

**Table 3 mps-02-00038-t003:** Demographic and clinical data collection form for patients on CPB.

**Data Collection form** **Extracorporeal Circulation Study** **(CPB)**	**Patient No.**Date:Time:Food intake:Water intake:
Age	Sex (M/F)	Weight (Kg)	Height (cm)
ICU Admission Diagnosis	
Hours on CPB	
Type of CPB	
Hypothermia	
Blood pump flow rate	
Type of membrane	
Serum bilirubin (µmol/L)	8 h Creatinine Clearance
Serum Creatinine (µmol/L)	
Serum Albumin (g/L)	
Total Proteins (g/L)	
Blood Urea (mmol/L)

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
