# Peer review of "Study Protocol for a Pilot, Open-Label, Prospective, and Observational Study to Evaluate the Pharmacokinetics of Drugs Administered to Patients during Extracorporeal Circulation; Potential of In Vivo Cytochrome P450 Phenotyping to Optimise Pharmacotherapy"

_mps, 2019, doi:10.3390/mps2020038_

Round 1

Reviewer 1 Report

The authors propose to use a standard drug probe cocktail to investigate the impact CPB and ECMO have on CYP450 metabolism in patients on these therapies vs a control set of patients undergoing laparoscopic or cholecystectomies.  It is understood that this manuscript is a protocol and does not contain any results from conduction of the study.

While this is an interesting concept there is one serious and several significant items that are not addressed in the protocol.

The serious flaw  relates to the complete disregard of how other meds the patients will be on in the study will impact the results.  Specifically meds that are substrates for the CYP450 enzymes being studied.  How will the authors account for observed alterations in metabolism as function of time and as compared to the control group for patients that will be on different meds that are substrates for the same enzymes as the cocktail probe substrates?  Using midazolam going to it hydroxyl metabolite via CYP3A4, how will the authors differentiate between the effect of CPB or ECMO vs other meds dosed that are CYP3A4 substrates?  What if the patients receives one of the cocktail drugs as a Rx med? Additionally the ECMO will go over several weeks meds may be discontinued or added during this time.  It does not seem like there is a way to adjust or interpret the data.  No mention was made of this in the protocol.  On this basis alone, without an explanation, it doesn't seem that the object of the study could be accomplished.

Significant concerns are the lack of any calculations showing the necessary number of patients to power the study to a defined level of significance. Seems the numbers listed (20 for CPB and 10 for ECMO) are bases more on census data than a required number for statistical significance.

Inclusion criteria does not mention any type of liver disease/dysfunction (hepatitis, cirrhosis nonalcoholic fatty liver disease, etc) which could have a significant impact on the results. Nor does it appear that this data will even be collected.

CPB and ECMO come in different doses/levels, it appears this data will be collected but  no mention is made as to how this will impact the results or be used.

In section 3.3 the dose rout of omeprazole is not stated.

It is stated that plasma samples will be kept for 7 years, is there any data to support the viability of samples for that long?

Author Response

Response to reviewer’s comments

Reviewer-1

1.              The serious flaw relates to the complete disregard of how other meds the patients will be on in the study will impact the results.  Specifically meds that are substrates for the CYP450 enzymes being studied.  How will the authors account for observed alterations in metabolism as function of time and as compared to the control group for patients that will be on different meds that are substrates for the same enzymes as the cocktail probe substrates?  Using midazolam going to it hydroxyl metabolite via CYP3A4, how will the authors differentiate between the effect of CPB or ECMO vs other meds dosed that are CYP3A4 substrates?  What if the patients receives one of the cocktail drugs as a Rx med? Additionally the ECMO will go over several weeks meds may be discontinued or added during this time.  It does not seem like there is a way to adjust or interpret the data.  No mention was made of this in the protocol.  On this basis alone, without an explanation, it does not seem that the object of the study could be accomplished.

Response: The overall activities that will be measured in the participants of the study is the combination of the impact of disease, genetics, environment and other medications that they receive. This is the major advantages of the CYP phenotyping over the other methods including genotyping.

Any changes during the ECMO treatment including changes in the medications or other interventions will be captured by the overall changes in the activities of the enzymes. Length of ECMO (like the length of cardiopulmonary bypass) will be considered as one of the variables that its impact on the activities of CYP enzymes will be investigated.

The CYP activities are measured using the ratio of the metabolite to the parent drug. If the participant receives one of the cocktail drugs as part of their medications, it does not affect the overall activities and the ratio of metabolite to the parent will be the same because all those cocktail substrates have linear pharmacokinetics and their metabolism won’t be affected by small changes in the concentrations of the substrates.

Response:

2.              Significant concerns are the lack of any calculations showing the necessary number of patients to power the study to a defined level of significance. Seems the numbers listed (20 for CPB and 10 for ECMO) are bases more on census data than a required number for statistical significance.

Response: This is a pilot and observational clinical study to test safety and feasibility of the protocol. CYP phenotyping study was never performed before in patients undergone cardiac surgery with cardiopulmonary bypass and ECMO. The calculations of the power of the study depends on the extent of variability in CYP activities in patients’ populations. The pilot study will provide us with an estimate of the variability and we use this information to determine the power of the subsequent larger cohort study in the future.

3. Inclusion criteria does not mention any type of liver disease/dysfunction (hepatitis, cirrhosis nonalcoholic fatty liver disease, etc) which could have a significant impact on the results. Nor does it appear that this data will even be collected.

Response: Clinicians will have access to the full list of pathological tests results before conducting the surgery or treating with ECMO and will decide if the patient is suitable for participating in the study.

Patients with serum bilirubin > 150 µmol/L have been excluded from the study. Additionally, we have added “Patients with liver disease / dysfunction” to the exclusion criteria.

3.              CPB and ECMO come in different doses/levels, it appears this data will be collected but no mention is made as to how this will impact the results or be used.

Response: Most of the patients recruited in this hospital will receive similar treatment in terms of the specifications of the CPB and ECMO treatment. The length of the treatment may be variable, and we can simply investigate the impact of the length of treatment on the changes in the CYP activities as one of the covariates. ECMO blood flow will be one of the covariates as it correlates well with severity of illness.

Similar doses of drugs in the phenotyping cocktail will be administered to CPB and ECMO patients. Phenotyping cocktail will be administered before and after cardiac surgery with CPB, where every patient will be the control for himself / herself. In the case of ECMO, phenotyping cocktail will be administered to the patient while on ECMO and after termination of ECMO to determine the effects of ECMO machine itself on the Cytochrome P450 activity. The comparison of these results will give us data about the impact of short-term (CPB) and long-term perfusion (ECMO) on CYP activity.

4.              In section 3.3, the dose route of omeprazole is not stated.

Response: “Omeprazole, 20 mg, one 20 mg tablet will be administered orally to the patient” is added.

5.              It is stated that plasma samples will be kept for 7 years, is there any data to support the viability of samples for that long?

Response: The storage period is a requirement for ethical aspect of the study and is very unlikely that samples need to be reanalysed after 7 years. However, some of the quality control (QCs) samples will be prepared when the study samples are analysed, and they will be stored in the same storage conditions as the study samples. Those QCs will be analysed later to make sure that the storage condition does not affect the analytes of interest. 

Reviewer 2 Report

The concept of the study sounds promising, but there are some suggestions for optimizing the protocol:

- why are you using regular doses - and not microdoses?

- why don’t you also genotype for CYP2C19, 2C9, 2D6?

- type of hemofilter, oxygenator and pumps will be recorded - please state how these data will be incorporated in the data analysis. 

Minor: 

- several typos - mainly inappropriate capitalization of nouns words

Author Response

Response to reviewers’ comments

Reviewer-2

1.              why are you using regular doses - and not microdoses?

Response: The single doses of relatively safe CYPs substrates that are used in this study are not expected to have any therapeutic/adverse effects. Substrates are used in this study with doses equal or smaller than previously CYP phenotyping cocktails. Those studies did not report any adverse effects related to the administration of the CYP substrates.

In addition, prescribing microdoses of CYP substrates can potentially cause some error during dispensing of the medications.

Details of the bioanalytical methods developed for quantification of OME, MDZ, DXM, LOS and their major metabolites for use with various CYP phenotyping cocktails in humans

Authors

Matrix

Calibration range (ng/mL)

Sample Volume

Dose of Drugs in the   cocktail (mg)

Puris et al., 2017

Serum

LOS: 0.05-250, EXP-3174: 0.25-250, OME:   0.025- 1000, 5-OH: 0.025-500, DXM: 0.005-100, DXT: 0.01-50, MDZ: 0.05-250,   1-HM: 0.05-1000

100 µL

LOS: 12.5, OME: 10, DXM: 30, MDZ: 1.85

Tanaka et al., 2014

Plasma

LOS: 1-1000, EXP-3174: 1-1000, OME:   1-1000, 5-OH: 1-1000, DXM: 0.2-100, DXT: 0.1-100, MDZ: 0.1-100, 1-HM: 0.1-100

300 µL

CAF: 100, LOS: 50, OME: 2, DXM: 30, MDZ: 1

de Andres et al., 2014

Plasma

LOS: 0.5-200, EXP-3174: 1-3000, OME:   0.5-1000, 5-OH: 0.5-1000, DXM: 1-100, DXT: 1-600

100 µL

OME : 20, LOS: 25, DXM: 30

Bosilkovska et al., 2014

Dried blood spots /   Plasma

OME: 0.4-200, 5-OH: 0.4-200, DXM: 0.2-200,   DXT: 0.5-500, MDZ: 0.1-100

1-HM: 0.2-200

50 µL

DXM: 10, OME: 5, MDZ: 1

Oh et al., 2012

Plasma

LOS: 0.1-40, EXP-3174: 0.1-40, OME:   0.05-20, 5-OH: 0.05-20, DXM: 0.008-0.8, DXT: 0.008-0.8, MDZ: 0.01-1, 1-HM:   0.04-4

450 µL

LOS: 2, OME: 0.2, DXM: 2, MDZ: 0.1

Ghassabian et al., 2009

Plasma

DXM: 3-400, DXT: 1.5-400,   OME: 7.8-1000, 5-OH: 7.8-1000, MDZ: 0.78-100, 1-HM: 0.78-100, LOS: 4-500,   EXP-3174: 4-500

200 µL

DXM: 30, OME: 20, MDZ: 2,   LOS: 25

Petsalo et al., 2008

Urine

DXM: 0.4-1000, DXT:   1-1000, OME: 0.4-1000, 5-OH: 0.4-2000, MDZ: 1-1000, 1-HM: 0.2-1000, LOS:   0.4-1000, EXP-3174: 0.4-1000

125 µL

LOS: 50, OME: 20, DXM: 25

Kumar et al., 2007

Pig Plasma

DXM: 10-2000, DXT:   10-2000, MDZ: 5-500, 1-HM: 5-500

200 µL

DXM: 0.5 mg/kg, MDZ: 0.25   mg/kg

Inje cocktail

Plasma

LLOQs: CAF: 5 ng/mL, PAX:   5 ng/mL, LOS: 5 ng/mL, EXP-3174: 5 ng/mL, OME: 5 ng/mL, 5-OH: 1.25 ng/mL,   DXM: 5 ng/mL, DXT: 5 ng/mL, MDZ: 0.1 ng/mL

500 µL

MDZ: 2, LOS: 30, CAF: 93,   OME: 20, DXM: 30

Yin et al 2004

Plasma

CAF: 50-5000,PAX:   50-5000, OME: 5-2500, 5-OH: 5-2500, MDZ: 1-100, 1-HM: 1-100

500 µL

MDZ: 3.75, CAF: 100, OME:   40

Chainuvati et al

Plasma

CAF: 0.25-250, PAX:   0.25-250, OME: 0.25-250, 5-OH: 0.25-250, MDZ: 0.25-250, 1-HM: 0.25-250, DXM:   0.25-250, DXT: 0.25-250

1 mL

CAF: 2 mg/Kg, DXM: 30,   OME: 40, MDZ: 0.025 mg/Kg

References:

Puris, E., et al., A liquid chromatography-tandem mass spectrometry analysis of nine cytochrome P450 probe drugs and their corresponding metabolites in human serum and urine. Anal Bioanal Chem, 2017. 409(1): p. 251-268.

Tanaka, S., et al., Simultaneous LC-MS/MS analysis of the plasma concentrations of a cocktail of 5 cytochrome P450 substrate drugs and their metabolites. Biol Pharm Bull, 2014. 37(1): p. 18-25.

de Andres, F., M. Sosa-Macias, and A. Llerena, A rapid and simple LC-MS/MS method for the simultaneous evaluation of CYP1A2, CYP2C9, CYP2C19, CYP2D6 and CYP3A4 hydroxylation capacity. Bioanalysis, 2014. 6(5): p. 683-96.

Bosilkovska, M., et al., Geneva cocktail for cytochrome p450 and P-glycoprotein activity assessment using dried blood spots. Clin Pharmacol Ther, 2014. 96(3): p. 349-59.

Oh, K.S., et al., High-sensitivity liquid chromatography-tandem mass spectrometry for the simultaneous determination of five drugs and their cytochrome P450-specific probe metabolites in human plasma. J Chromatogr B Analyt Technol Biomed Life Sci, 2012. 895-896: p. 56-64.

Ghassabian, S., et al., A high-throughput assay using liquid chromatography-tandem mass spectrometry for simultaneous in vivo phenotyping of 5 major cytochrome p450 enzymes in patients. Ther Drug Monit, 2009. 31(2): p. 239-46.

Petsalo, A., et al., Analysis of nine drugs and their cytochrome P450-specific probe metabolites from urine by liquid chromatography-tandem mass spectrometry utilizing sub 2 microm particle size column. J Chromatogr A, 2008. 1215(1-2): p. 107-15.

Kumar, A., H.J. Mann, and R.P. Remmel, Simultaneous analysis of cytochrome P450 probes-dextromethorphan, flurbiprofen and midazolam and their major metabolites by HPLC-mass-spectrometry/fluorescence after single-step extraction from plasma. J Chromatogr B Analyt, 2007 Jun 15;853(1-2):287-93.

Ryu, J.Y., et al., Development of the "Inje cocktail" for high-throughput evaluation of five human cytochrome P450 isoforms in vivo. Clin Pharmacol Ther, 2007. 82(5): p. 531-40.

Yin, O.Q., et al., Rapid determination of five probe drugs and their metabolites in human plasma and urine by liquid chromatography/tandem mass spectrometry: application to cytochrome P450 phenotyping studies. Rapid Commun Mass Spectrom, 2004. 18(23): p. 2921-33.

Chainuvati, S., et al., Combined phenotypic assessment of cytochrome p450 1A2, 2C9, 2C19, 2D6, and 3A, N-acetyltransferase-2, and xanthine oxidase activities with the "Cooperstown 5+1 cocktail". Clin Pharmacol Ther, 2003. 74(5): p. 437-47.

2.              why don’t you also genotype for CYP2C19, 2C9, 2D6?

Response: Genotyping is not able to show the impact of disease, environment, other medications on the activities of CYP enzymes. In this study, the activities of CYPs in each patient is compared with “before the intervention” and the genotyping parameters are the same between before and after intervention. In terms of comparison among CPB, ECMO and control groups, the number of patients is not big enough to show the genotyping differences between study and control groups.

3.              type of hemofilter, oxygenator and pumps will be recorded - please state how these data will be incorporated in the data analysis. 

Response: Being a single centre study, all our CPB, ECMO and renal replacement therapy equipment are standardised. Equally, the intra and post-operative management practices are also largely standardised. Any variation will be documented and accounted for.

4.         Several typos – mainly inappropriate capitalization of noun words.

Response: Typo errors have been rectified in the latest manuscript document.

Reviewer 3 Report

Sreevatsav et al propose a clinical study to evaluate the effect of ECMO on cytochrome P450 gene regulation in humans. The study is sound, the protocol is well written, and subsequent findings should be of interest to readers. 

Specific Comments:

1) One primary concern with this protocol is that it uses oral dosing for substrates. This introduces a level of variability (absorption) to the method. Seems clear that surgical patients will be fasted. Will the patients be fasted and treated similarly on other days (day, time, food intake?). Will the participants be given a similar amount of water? The dosing protocol is not well described. It seems the treatment protocol should be well thought through to minimize variability in collected data.

2) Some of these substrates are also inducers of P450s (omeprazole will induce CYP1As). I don't know if they will induce over several hours, but certainly could be elevated after repeated dosings. A good control group for this is necessary. 

3) Are the authors concerned that some groups of samples will be drawn from venous catheter while blood for other groups will be obtained from peripheral arterial lines. Are these sources of blood similar with regard to substrate levels?

4) P450s are regulated by circadian rhythm. Will the participants receive the substrate dosing at the same time each day? Will that time of day be the same on the day of surgery vs after recovery? 

Author Response

Response to reviewer’s comments

Reviewer-3

1.              One primary concern with this protocol is that it uses oral dosing for substrates. This introduces a level of variability (absorption) to the method. Seems clear that surgical patients will be fasted. Will the patients be fasted and treated similarly on other days (day, time, food intake?). Will the participants be given a similar amount of water? The dosing protocol is not well described. It seems the treatment protocol should be well thought through to minimize variability in collected data.

Response: Very good point. We make sure that we record all the details of food and water intake in patients. The protocol has been updated to capture those details. Table-2 & 3 have been modified accordingly.

2.              Some of these substrates are also inducers of P450s (omeprazole will induce CYP1As). I don't know if they will induce over several hours, but certainly could be elevated after repeated dosing. A good control group for this is necessary. 

Response: Only single dose of omeprazole (and other CYP substrates) will be administered during each stage of CYP phenotyping and induction of CYP enzymes with this dose (20 mg) is very unlikely. One of the parameters that have been determined during the validation of this CYP phenotyping cocktail (Yin et al 2005) was the lack of pharmacokinetic and pharmacodynamic interactions among the substrates.  In means that the administration of omeprazole alone, or in combination with 4 other substrates did produce similar CYP2C19 activities in participants.

Reference:

Yin, O.Q., et al., Rapid determination of five probe drugs and their metabolites in human plasma and urine by liquid chromatography/tandem mass spectrometry: application to cytochrome P450 phenotyping studies. Rapid Commun Mass Spectrom, 2004. 18(23): p. 2921-33.

3.              Are the authors concerned that some groups of samples will be drawn from venous catheter while blood for other groups will be obtained from peripheral arterial lines. Are these sources of blood similar with regard to substrate levels?

Response: The concentrations of the substrates should be similar in both peripheral arterial and venous blood.

4.              P450s are regulated by circadian rhythm. Will the participants receive the substrate dosing at the same time each day? Will that time of day be the same on the day of surgery vs after recovery? 

Response: While we appreciate changes in the CYP activities due to circadian rhythm, those changes must be minimum compared with changes caused by the surgical interventions. It will be very challenging to do the CYP phenotyping procedure at the same time of the day for all patients.  

Round 2

Reviewer 1 Report

The authors addressed many of the items previously mentioned, however it still seems that based on the study design there are too many variables (as previously mentioned) to get meaningful data from the study (including how to power a full study).  The observed changes in metabolism of the probe cocktail compounds even using the ratio of metabolite to parent drug will not work if the enzymes are saturated.  This could be the case depending on other Rx meds the patients are on.

Author Response

New comment:

"The authors addressed many of the items previously mentioned, however it still seems that based on the study design there are too many variables (as previously mentioned) to get meaningful data from the study (including how to power a full study).  The observed changes in metabolism of the probe cocktail compounds even using the ratio of metabolite to parent drug will not work if the enzymes are saturated.  This could be the case depending on other Rx meds the patients are on."

As was mentioned before, this is the first pilot and observational study in two patients’ populations with the aim to provide information regarding the safety, feasibility and variability in the activities of CYP enzymes. The number of participants to be recruited is based on the expected number of admitted patients in the hospital in the period of the study. If we see high variability in the CYP activities which does not provide enough information for the calculation of the power of the subsequent study, we need to extend the study to recruit more patients (this has been added to the manuscript).

These patients are unlikely to receive Rx meds that are strongly induce or inhibit CYP enzymes because it can adversely affect the efficacy and safety of their current medications disregarding of their participation in the study. However, to rule out the possibility of significant clinical interaction, we have added another exclusion criteria “Using drugs that are known to be strong inhibitors or inducers of CYP enzymes (Flockhart table)” and also “smokers” as it is known to induce CYP1A2 activity.